# DiLoCo: Distributed Low-Communication Training of Language Models

Arthur Douillard [* 1]   Qixuan Feng [* 1]   Andrei A. Rusu [* 1]   Rachita Chhaparia [1]   Yani Donchev [1]
Adhiguna Kuncoro [1]   Marc'Aurelio Ranzato [1]   Arthur Szlam [1]   Jiajun Shen [1]

## Abstract

Large language models (LLM) have become a critical component in many applications of machine learning. However, standard approaches to training LLM require a large number of tightly interconnected accelerators, with devices exchanging gradients and other intermediate states at each optimization step. While it is difficult to build and maintain a single computing cluster hosting many accelerators, it might be easier to find several computing clusters each hosting a smaller number of devices. In this work, we propose a distributed optimization algorithm, Distributed Low-Communication (DiLoCo), that enables training of language models on islands of devices that are poorly connected. The approach is a variant of federated averaging, where the number of inner steps is large, the inner optimizer is AdamW, and the outer optimizer is Nesterov momentum. On the widely used C4 dataset, we show that DiLoCo on 8 workers performs as well as fully synchronous optimization while communicating 500 times less. DiLoCo exhibits great robustness to the data distribution of each worker. It is also robust to resources becoming unavailable over time, and vice versa, it can seamlessly leverage resources that become available during training.

## 1. Introduction

Language models have shown remarkable ability to generalize to new tasks, and are at the heart of a multitude of new applications of machine learning. Because performance has scaled with model size, practitioners train increasingly larger models on increasingly large data. Nevertheless, at a high level, the basic training approach remains standard mini-batch back-propagation of the error.

At modern scale, training via standard back-propagation poses unprecedented engineering and infrastructure challenges. To start, several thousands of devices need to be powered and be placed at the same physical location; and interconnected with high-bandwidth cables to minimize latency. Careful software engineering is required to orchestrate the passage of gradients, parameters and intermediate states between these devices at each optimization step, keeping all devices fully utilized. Furthermore, the more devices that are used for each synchronous training step, the more chances there are that one of them fails, risking halting training, or introducing subtle numerical issues. Moreover, the current paradigm poorly leverages heterogeneous devices, that might have different speed and topology. In the simplest terms, it is difficult to co-locate and tightly synchronize a large number of accelerators.

In this work, we take inspiration from literature on Federated Learning, to address the above mentioned difficulties. In Federated Learning, there are $k$ workers, each operating on their own island of devices, each consuming a certain partition of the data, and each updating a model replica. Such workers perform some amount of work locally, and then exchange gradients every $H$ steps to get their model replica back in sync.

We propose a variant of the popular Federated Averaging (FedAvg) algorithm (McMahan et al., 2017), or a particular instantiation with a momentum-based optimizer as in the FedOpt algorithm (Reddi et al., 2021), whereby the number of inner steps is large, the inner optimizer is replaced with AdamW, and the outer optimizer with Nesterov Momentum for best performance. This combination enables us to address the shortcomings mentioned above, because a) while each worker requires co-located devices their number is roughly $k$ times smaller than the total, b) workers need not to communicate at each and every single step but only every $H$ steps which can be in the order of hundreds or even thousands, and c) while devices within an island need to be homogeneous, different islands can operate with different device types. We dub this approach Distributed Low-Communication (DiLoCo) training.

In a large-batch training setting with overtraining, our empirical validation on the C4 dataset (Raffel et al., 2020)

---
*Equal contribution [1]Google DeepMind. Correspondence to: Arthur Douillard <douillard@google.com>, Marc'Aurelio Ranzato <ranzato@google.com>.

Accepted to the Workshop on Advancing Neural Network Training at International Conference on Machine Learning (WANT2024).

demonstrates that DiLoCo can achieve even better performance (as measured in perplexity) than a fully synchronous model, while communicating 500 times less. DiLoCo is capable of effectively utilizing several islands of devices at training time, despite a low bandwidth connectivity among these islands. Finally, at inference time the resulting model has the same size and speed as a model trained in fully synchronous mode.

Our experiments show that DiLoCo is robust against different data distributions used by local workers and frequency of global parameter updates. Finally, DiLoCo exhibits robustness to island failure, and nicely leverage islands whenever these become available.

## 2. DiLoCo

We assume that we have a base model architecture (e.g., a transformer) with parameters $\theta$. We denote a training dataset as $\mathcal{D} = \{(\mathbf{x}, \mathbf{y}), ...\}$ with $\mathbf{x}$ and $\mathbf{y}$ being respectively the input data and target. In language modeling (Vaswani et al., 2017), the input is a sequence of tokens and the target is the input sequence shifted by one. When the dataset is split across multiple shards, we denote the $i$-th shard with $\mathcal{D}_i$.

DiLoCo training proceeds as outlined in Algorithm 1 (Reddi et al., 2021), and illustrated in Figure 1. First, we start from an initial model with parameters $\theta^{(0)}$, which can be initialized at random or using a pretrained model (see subsection 3.1). We also have $k$ workers each capable of training a model replica and $k$ shards of data, one for each worker.

There are two optimization processes. There is an *outer* optimization (line 1, 12, and 14 in Algorithm 1), which consists of $T$ outer steps. At each outer step $t$, gradients from each worker are gathered, averaged and sent to an outer optimizer (OuterOpt) to update the shared copy of the parameters. Afterwards, this shared copy of the parameters is re-dispatched to each local worker (line 3).

Within each phase, each worker (line 3) performs *independently and in parallel* its own inner optimization (lines 4 to 9) for $H$ steps using an inner optimizer, denoted by InnerOpt. Each worker samples data from its own shard (line 5), and updates its own local copy of the parameters (line 8). Note that the inner optimization consists of $H \gg 1$ steps; for instance, several hundred steps. Therefore, communication across workers is minimal.

Once all workers have completed their inner optimization step, the cumulative gradients of each worker are averaged (line 12), and the resulting *outer gradient* is used to update the shared copy of the parameters (line 14), which is then used as starting point for the next round of inner optimizations. This is the only time when communication among

workers is required, and it happens once every $H$ inner optimization steps. In total, a worker trains for $N = T \times H$ inner steps.

In our work, we use as *inner* optimizer (InnerOpt) AdamW (Kingma & Ba, 2014; Loshchilov & Hutter, 2019), which is the most widely used optimizer for transformer language models. Note that the vast majority of the literature usually use SGD instead, we found it to be inefficient for training transformers. As for the *outer* optimizer (OuterOpt) we use Nesterov momentum (Sutskever et al., 2013) because it gave the best convergence empirically (see Figure 6). When OuterOpt is SGD, then the outer optimizer is equivalent to classical Federated Averaging (McMahan et al., 2017). If the total number of outer optimization steps $T$ is further set to 1, then DiLoCo reduces to "souping" (Wortsman et al., 2021). Finally, if the number of inner optimization steps $H$ is set to 1 and InnerOpt is SGD, DiLoCo is equivalent to large-batch training with data-parallelism.

Overall, DiLoCo can be interpreted as a data parallelism method that requires very little communication, and therefore, it can scale to workers that are poorly connected, e.g., workers placed in very distant geographic regions. Workers could of course use standard data and model parallelism for their inner optimization.

---

**Algorithm 1** DiLoCo / FedOpt Algorithm

---

**Require:** Initial model $\theta^{(0)}$
**Require:** $k$ workers
**Require:** Data shards $\{\mathcal{D}_1, \ldots, \mathcal{D}_k\}$
**Require:** Optimizers InnerOpt and OuterOpt
1: **for** outer step $t = 1 \ldots T$ **do**
2:      **for** worker $i = 1 \ldots k$ **do**
3:          $\theta_i^{(t)} \leftarrow \theta^{(t-1)}$
4:          **for** inner step $h = 1 \ldots H$ **do**
5:              $x \sim \mathcal{D}_i$
6:              $\mathcal{L} \leftarrow f(x, \theta_i^{(t)})$
7:                       ▷ Inner optimization:
8:              $\theta_i^{(t)} \leftarrow \texttt{InnerOpt}(\theta_i^{(t)}, \nabla_{\mathcal{L}})$
9:          **end for**
10:      **end for**
11:              ▷ Averaging outer gradients:
12:      $\Delta^{(t)} \leftarrow \frac{1}{k} \sum_{i=1}^{k} (\theta^{(t-1)} - \theta_i^{(t)})$
13:              ▷ Outer optimization:
14:      $\theta^{(t)} \leftarrow \texttt{OuterOpt}(\theta^{(t-1)}, \Delta^{(t)})$
15: **end for**

---

## 3. Experiments

In this section we report the main experiments validating DiLoCo. We consider a language modeling task on the C4 dataset, a dataset derived from Common Crawl (Raffel

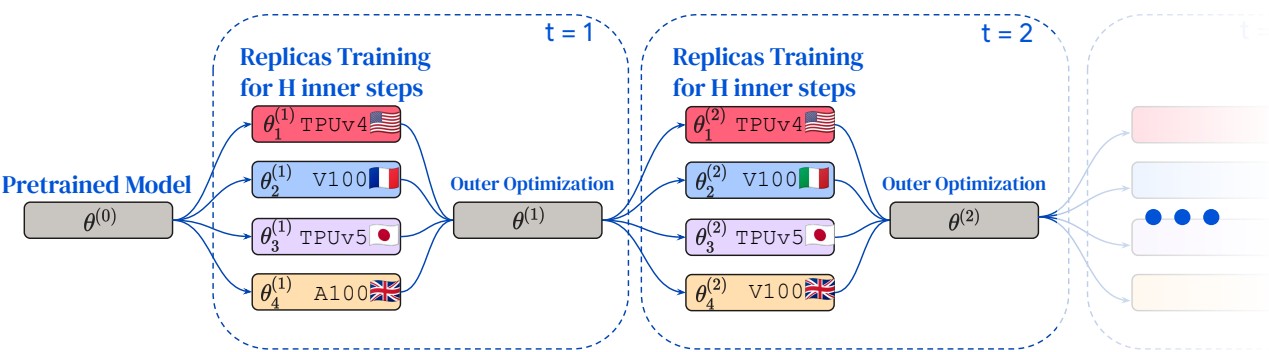

*Figure 1.* **DiLoCo**: First, a pretrained model $\theta^{(0)}$ is replicated $k$ times (in this illustration $k = 4$) and each worker $\theta_i^{(1)}$ trains a model replica on its own shard of data for $H$ steps independently and in parallel. Afterwards, workers average their outer gradients and an outer optimizer updates the global copy of the parameters $\theta^{(1)}$. This will then be re-dispatched to the workers. The process repeats $T$ times (in this illustration only the first two iterations are displayed). Each replica can be trained in different locations of the world, with different accelerators.

*Table 1.* **Model Configuration** for the three evaluated sizes. All are based on the transformer architecture, chinchilla-style (Hoffmann et al., 2022).

| Hyperparameter | 60M | 150M | 400M |
|---|---|---|---|
| Number of layers | 3 | 12 | 12 |
| Hidden dim | 896 | 896 | 1536 |
| Number of heads | 16 | 16 | 12 |
| K/V size | 64 | 64 | 128 |
| Vocab size | | 32,000 | |

et al., 2020). We report perplexity on the validation set against number of steps used at training time, which is a good proxy for wall clock time since communication across workers is rather infrequent. The total number of steps is set to 88,000. We consider three model sizes, all decoder-only transformers adapted from the Chinchilla architecture (Hoffmann et al., 2022). Their respective configuration is described in Table 1. We perform experiments both in the i.i.d. and non-i.i.d. settings, meaning when the data distribution of the shards $\mathcal{D}_i$ is the same for all $i$ and when these are different like in heterogeneous federated learning. Since the latter is a more challenging use case, we use this setting by default except when indicated otherwise. Similarly, by default all training experiments start from a transformer language model pretrained for 24,000 steps on the same training set, refer to subsection 3.1 for further details.

In our experiments we have searched over the hyper-parameters of the outer optimizer (*e.g.* learning rate, momentum, etc.). We use a sequence length of 1,024 tokens and a batch of size 512 but otherwise we left unchanged the inner optimization and model architecture. We list all the

hyper-parameters in the appendix (Table 5).

In Figure 2, we show the performance through time of DiLoCo (in blue with $k = 8$ replicas in the non-i.i.d. data setting) when each worker performs $T = 128$ times $H = 500$ inner steps (64,000 steps in total). In this experiment, DiLoCo starts from a model $\theta^{(0)}$ pretrained for 24,000 steps.

There are four baselines. The first baseline (1) is a model trained from scratch for 88,000 steps (in red), the second (2) starts from a model pretrained for 24,000 steps and performs an additional 64,000 steps (in teal). The third baseline (3) starts from the same pre-trained model, but during finetuning uses an $8\times$ bigger batch size (in purple) with microbatching (also called *gradient accumulation*). The fourth baseline (4) is running the standard batch size for $8\times$ the number of updates. Finally the last row is our model DiLoCo (5). We compare in Table 2 all baselines with respect to the communication cost, time spent training, and the amount of compute & data used. Increasing the batch size can be done in two manners: with data parallelism (second row) at the cost of increased communication, or with microbatching (third row) at the cost of longer training time. DiLoCo (last row) doesn't increase training time, communicates $H = 500\times$ less than the second baseline (and is thus amenable to distributed training across compute islands), while also reaching better generalization. Increasing by $8\times$ the number of updates improves perplexity over our method, but at the cost of being $8\times$ slower.

### 3.1. Ablations

In the following section, we perform extensive ablations of DiLoCo to better understand its capabilities and stress-test its limits. More results are also available in the appendix.

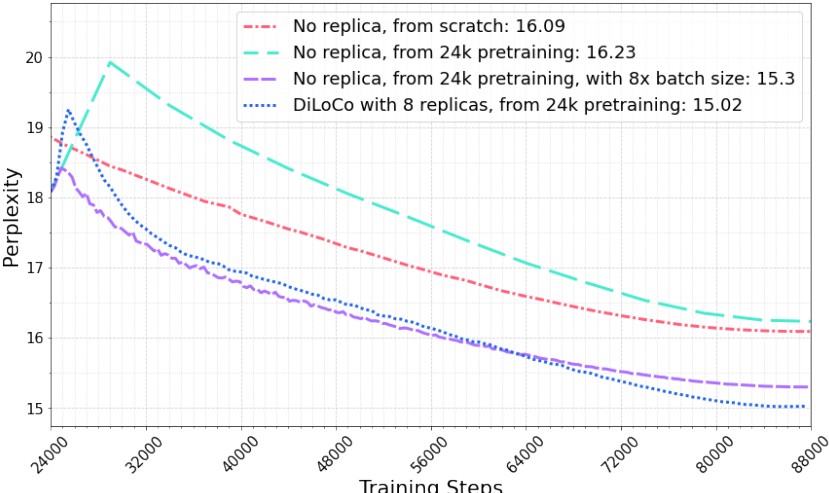

*Figure 2.* **Main result**: After pretraining a 150M baseline for 24,000 training steps on C4, we compare networks finetuned for an additional 64,000 steps (teal using the same batch size, and purple using 8 times bigger batch size), and a transformer model trained from scratch (red). DiLoCo(blue) using 8 workers yields lower perplexity, even compared to the baseline using 8 times bigger batch size, while being 8 times faster in wall-clock time and communicating 500 times less.

*Table 2.* **Trade-offs of various training algorithms**: We compare four baselines *vs* DiLoCo across their communication cost, time spent, and compute & data used. For the same time and amount of compute, we can compare the second baseline and DiLoCo. The former communicates gradients at each time step ($N$ total steps), while DiLoCo communicates $H = 500$ times less (and is amenable to distributed training) while also reaching better generalization performance. Note that $T = N/H$ (see Algorithm 1). Also note that microbatching is sometimes called *gradient accumulation*.

| Model | Parallel (P) vs Serial (S) | Communication | Time | Compute & Data | Perplexity |
|---|---|---|---|---|---|
| 1) Baseline | S | 0 | $1\times$ | $1\times$ | 16.23 |
| 2) Baseline, $8\times$ batch size with data parallelism | P | $8 \times N$ | $1\times$ | $8\times$ | 15.30 |
| 3) Baseline, $8\times$ batch size with microbatching | S | 0 | $8\times$ | $8\times$ | 15.30 |
| 4) Baseline, $8\times$ updates | S | 0 | $8\times$ | $8\times$ | 14.72 |
| 5) DiLoCo | P | $8 \times N/H$ | $1\times$ | $8\times$ | 15.02 |

**Number of Pretraining Steps** For all experiments here we perform 88,000 training steps. A subset of those steps are done during the pretraining stage, and the remainder with DiLoCo. In Figure 3, we study the impact of the number of pretraining steps on the final generalization performance in a non-i.i.d. data regime. Specifically, we compare no pretraining (in teal), pretraining of 12k (in purple), 24k (in red), and 48k (in orange) steps. We highlight the pretrain's ending and DiLoCo's beginning with vertical dashed lines. Note that as we keep the total amount of steps (wall-clock time) fixed, few or no pretraining steps will result in more compute spent overall.

In general, we observe that starting DiLoCo before 24k steps achieves a similar final PPL, demonstrating the robustness of the approach. Interestingly, performance is not degraded even when starting from a randomly initialized network. This result contradicts the findings of prior work on *post local-SGD* (Lin et al., 2020) and its large-scale study on a vision classification task (Ortiz et al., 2021).

The attentive reader may also note spikes in perplexity after the vertical dashed lines: a warm-up of the inner learning rate is the culprit. Despite the transient spike, such warm up is ultimately beneficial, as previously noted also in the *continual pretraining* setting by Gupta et al. (2023).

**Communication frequency** In order to scale up distributed training across a set of poorly connected machines, the frequency of communication needs to be reduced. Doing a single communication at the training's end (Wortsman et al., 2022a) is sub-optimal. Most works instead consider communicating every $H \leq 20$ steps (Ortiz et al., 2021), which is too frequent for many distirbuted learning applications.

In Figure 4, we vary the communication frequency for a 150M transformer, in the non-i.i.d. data regime, from

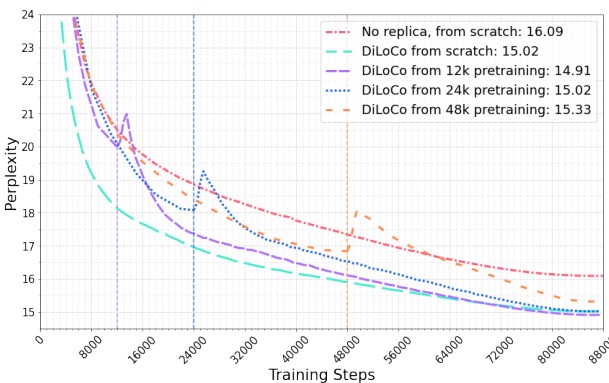

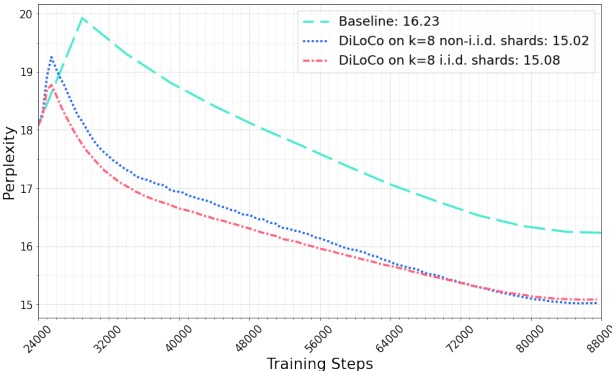

*Figure 3.* **Impact of number of pretraining steps** in a non-i.i.d. setting. DiLoCo can be initialized from a pretrained model $\theta^{(0)}$, or even from scratch with minimal (-0.1 PPL) degradation of model quality. The vertical dashed lines indicate the transition between pretraining and DiLoCo training.

*Figure 5.* **i.i.d. *vs* non-i.i.d. data regimes**: DiLoCo converges faster in the i.i.d. setting but towards the end both data regimes attain similar generalization, highlighting the robustness of DiLoCo.

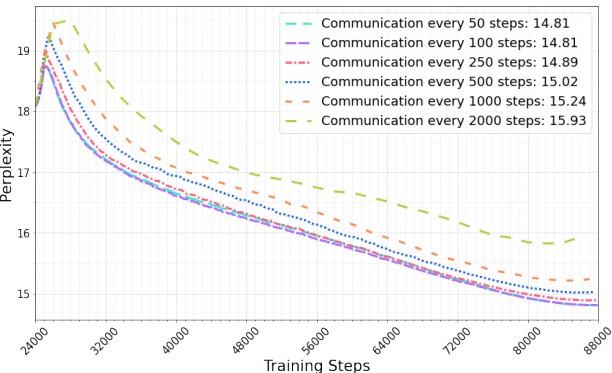

*Figure 4.* **Varying the communication frequency** every $H = \{50, 100, 250, 500, 1000, 2000\}$ steps in a non-i.i.d setting.

setting by clustering with $k$-Means the entire training set using a pretrained model's last layer features. The i.i.d. setting is a random partitioning of the data. We showcase in Figure 5 the performance of DiLoCo with $k = 8$ workers/shards in a non-i.i.d. setting (in blue) and i.i.d setting (in red). Despite the latter converging faster early on in training, the final generalization performance of the two settings is comparable. Intuitively, we would expect the non-i.i.d. setting to yield worse performance because each worker might produce very different outer gradients, but DiLoCo exhibits very strong robustness. The reason why this might be happening is further investigated in the appendix (subsection A.2).

$H = 50$ steps (in teal) to $H = 2000$ steps (in green). In general, we observe that communicating more frequently improves generalization performance. However, communicating more frequently than $H = 500$ steps leads to diminishing returns. Moreover, the performance degradation is very mild up to $H = 1000$ steps. For instance, when $H = 1000$ the perplexity increases by only $2.9\%$ relative to $H = 50$, despite communicating $20\times$ less. Based on these considerations, for all remaining experiments we choose $H = 500$ as this strikes a good trade-off between generalization performance and communication cost.

**i.i.d. vs non-i.i.d. data regimes** According to Gao et al. (2022), the distribution of the data across replicas can have a significant impact on generalization. In this ablation study we assess the effect that different data distributions have on the convergence of DiLoCo.

Similarly (Gururangan et al., 2023), we create the non-i.i.d.

**Number of replicas** We now investigate the impact of the number of replicas/clusters in Table 3, assuming there are as many workers as there are shards of data. The results in Table 3 show that increasing the number of replicas improves generalization performance, but with diminishing returns when there are more than 8 workers. This finding applies to both i.i.d. and non-i.i.d. settings. Unlike what is reported in prior work in the vision domain on ImageNet (Ortiz et al., 2021), we do not observe significant performance degradation by increasing the number of replicas.

**Model size** In Table 4 we vary the model size. We train models of size 60, 150 and 400 million parameters. We consider the usual setting where data distribution is non i.i.d. and all workers start from a model (of the same size) pretrained for 24,000 steps. Hyper-parameters were tuned on the 150M model, which may be sub-optimal for the other model sizes. We observe a monotonic improvement of performance as the model size increases. We surmise that (1) in an overtrained setting with large amount of steps, larger models are more efficient at fitting the same amount of data (Nakkiran et al., 2019), and (2) as the linear connectivity literature (Ilharco et al., 2022) suggests, larger models are

*Table 3.* **Impact of the number of replicas/clusters** on the evaluation perplexity for a fixed amount of inner steps per replica (150M parameters each). With more replicas, the model consumes more data and uses more compute overall, although this requires very infrequent communication (once every 500 inner steps).

| Number of replicas | *i.i.d* | non-*i.i.d* |
|:---:|:---:|:---:|
| 1 | 16.23 | |
| 4 | 15.23 | 15.18 |
| 8 | 15.08 | 15.02 |
| 16 | 15.02 | 14.91 |
| 64 | 14.95 | 14.96 |

*Table 4.* **Varying the model size**: For each model size, we report the relative and absolute perplexity (PPL) improvements of DiLoCo over the baseline (using a single worker). DiLoCo uses 8 workers and non-i.i.d. shards.

| Model Size | Relative (%) | Absolute (PPL) |
|:---:|:---:|:---:|
| 60M | 4.33% | 1.01 |
| 150M | 7.45% | 1.21 |
| 400M | 7.49% | 1.01 |

less subject to interference when averaging their parameters.

**Outer Optimizers**  We experimented with various outer optimizers (see L14 of Algorithm 1). For each, we tuned their momentum if any, and their outer learning rate. We found that using as outer optimizer SGD (equivalent to FedAvg (McMahan et al., 2017)) or Adam (eq. to FedOpt (Reddi et al., 2021)) performed poorly, as shown in Figure 6. Adam was particularly unstable with a high second order momemtum norm. We alleviated the issue by increasing the $\epsilon$ factor to 0.1. We found Nesterov optimizer (Sutskever et al., 2013) (see FedMom in (Huo et al., 2020)) to perform the best. In particular, the setting with outer learning rate equal to 0.7 and outer momentum equal to 0.9 is very robust, and it is adopted for all our experiments throughout. We hypothesize that the Nesterov's gradient correction is particularly helpful with the outer gradient that span hundred of training steps.

We also considered decaying the outer learning rate with a cosine scheduling but it resulted in similar performance. Since we decay the inner learning rate, the outer gradient norm gets naturally smaller over the course of training, removing the need to further decay the outer learning rate.

**Adaptive compute pool**  The total amount of compute any given user has, is rarely constant over time. For instance, a preemptible machine, that is regularly killed, is a cheaper alternative to a dedicated server. Similarly, university's com-

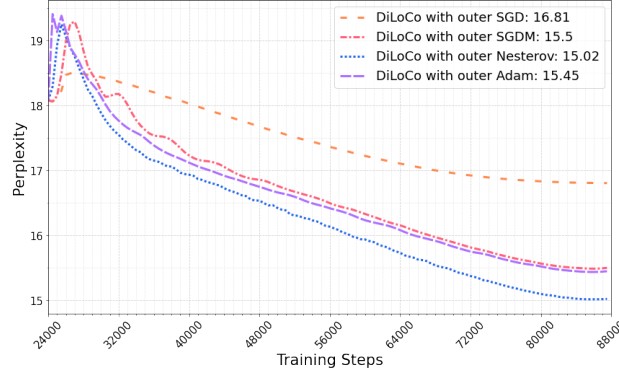

*Figure 6.* **Outer Optimizers**: Comparison of outer optimizers.

puting clusters often use *karma* systems to balance compute among all users, but this means that resources available to each user vary over time. Finally, a collaborative system like Petals (Borzunov et al., 2022) or (Diskin et al., 2021) where individual users provide their own devices to the shared compute pool is subject to extreme pool resizing depending on how many people participate at any given time.

In this study, we then explore the performance of DiLoCo when the amount of compute varies throughout training. In our case, the amount of compute is varied by changing the number of replicas used in an i.i.d. setting. In Figure 7, we show the validation perplexity through time when using different schedules of compute allocation. `Constant local` (in green) and `Constant Distributed` (in blue) use a constant amount of replicas: respectively 1 (baseline) and 8 (standard DiLoCo setting). `Doubling Compute` (in teal) and `Halving Compute` (in purple) use respectively 4 and 8 replicas during the first half of the training, and then 8 and 4. `Ramping Up` (in red) and `Ramping Down` (in orange) ramps up (respectively ramps down) the compute from 1 to 8 (resp. from 8 to 1).

We observe that the factor determining the ultimate generalization ability of the model is the total amount of compute given to DiLoCo, but this is robust to how the budget is spread over time. For instance, `Doubling Compute` and `Halving Compute` use as much compute in total and achieve similar performance. Similarly, `Ramping Up` and `Ramping Down` obtain similar performance despite the different budgeting schedule, and their generalization is worse than other baselines using more total compute. In conclusion, models quality is affected by the total amount of compute, but not as much by how such computed is allocated over time.

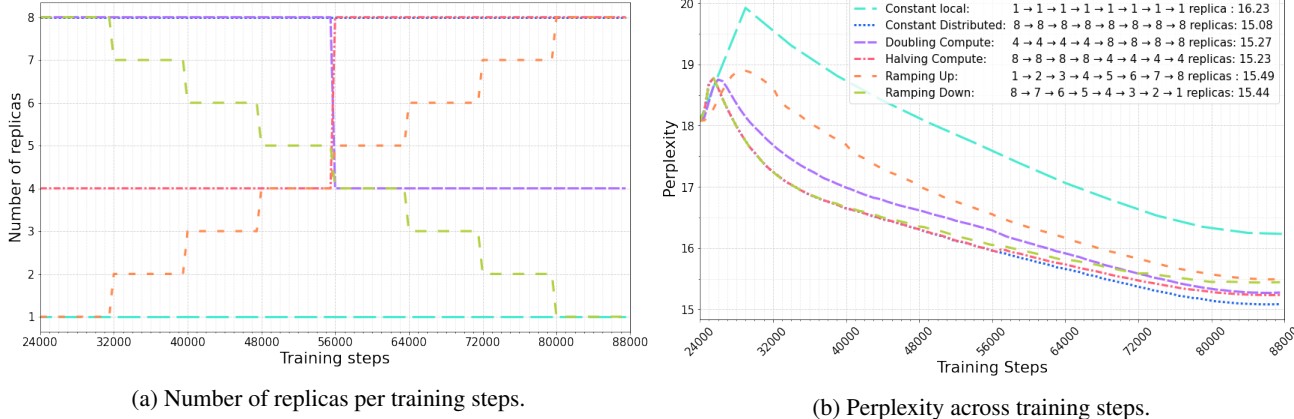

(a) Number of replicas per training steps.

(b) Perplexity across training steps.

*Figure 7.* **Adaptive compute**: We vary the number of replicas (i.e., the amount of compute) across time. Models generalize equally well for the same total amount of compute, regardless of how this is made available over time.

## 4. Related Work

In this section we review relevant work from the literature, limiting the discussion to only few representative works given the large body of literature.

We cover the literature of distributed learning, specifically local SDG and federated learning. We also relate to recent works done on linear mode connectivity which inspired much of our work.

### 4.1. Local SGD and Federated Learning

Several communities have proposed and studied local SGD. To the best of our knowledge, the first instantiation was in (McMahan et al., 2017) who introduced the concept of federated learning and local SGD as a way to enable learning on a network of mobile devices which retain private access to their own data. In this work, the outer optimization consists of a mere parameter averaging step. This was later extended to more powerful outer optimizers by (Wang et al., 2020; Reddi et al., 2021); this work inspired our use of Nesterov momentum in the outer optimization. (Lin et al., 2020) considered local SGD as a way to improve generalization when learning with large batch sizes. (Stich, 2019) instead focused on local SGD because of its ability to limit communication in distributed learning, a perspective we share also in our work. To the best of our knowledge, only FedMom (Huo et al., 2020) considers Nesterov as the outer optimizer as we did. While they also tackle a language modeling task, the setting is much smaller (1-layer LSTM), with only 2 replicas, and rather frequent communication (every 20 inner steps). In our work instead, we consider a larger setting with up to a 400M transformer language model, across up to 64 replicas, and up to $100\times$ less communication. Furthermore, we use AdamW as inner optimizer while they used SGD.

(Ortiz et al., 2021) is one of the few works in federated learning / local sgd body of literature that has validated on a large-scale setting. They consider ImageNet (Deng et al., 2009) with Resnet50 and Resnet101 (He et al., 2015), and found that local sgd struggles at scale. In particular, they reported that fewer inner steps (e.g., $H = 8$), no pretraining, and a relatively large number of replicas ($\geq k = 16$) degrade generalization. Thus the authors conclude that "*local SGD encounters challenges at scale.*". Instead, we show in section 3 that DiLoCo can robustly operate while communicating $125\times$ less ($H = 1000$), even without pretraining, and using up to $4\times$ more replicas ($k = 64$) both in the i.i.d. and non-i.i.d. settings. Recently, multiple works (Presser, 2020; Diskin et al., 2021; Ryabinin et al., 2021) also applied Local SGD for language models but without outer optimization.

### 4.2. Linear Mode Connectivity

The field of linear mode connectivity studies how to linearly interpolate between several models in parameters space, to yield a single model with the best capabilities of all models combined (Frankle et al., 2020; Wortsman et al., 2021). A surprising result from this field is the relative easiness to find a linear interpolation between several models where all intermediary points have a low loss, avoiding any *loss barrier*. Specifically, (Wortsman et al., 2022c) started from a pretrained model, finetuned different replicas on various tasks or choice of hyperparameters (Wortsman et al., 2022b), and then averaged the resulting parameters. Originally proposed in the vision domain, this method has then been used also in NLP (Li et al., 2022), RLHF (Ramé et al., 2023a), noisy data (Rebuffi et al., 2022), and OOD (Ramé et al., 2023b). Recently, several works studied other ways to alleviate loss barriers (Jordan et al., 2023; Stoica et al., 2023; Jin et al., 2023). While we didn't apply any of these methods to DiLoCo, they are complementary and could be used in

future works.

The majority of works on linear connectivity considers only averaging once all replicas have been fully finetuned, while we exploit the linear mode connectivity *during* training. There are however notable exceptions: BTM (Li et al., 2022) and PAPA (Jolicoeur-Martineau et al., 2023) are roughly equivalent to our framework but use as outer optimizer `OuterOpt = SGD(lr=1.0)`. The former communicates very little because each replica is fully finetuned on a task before synchronization. The latter communicates every few steps and with at most 10 replicas. Finally, (Kaddour, 2022) only considers a few previous checkpoints of the same model trained on a single task, and don't re-use it for training. Git-theta (Kandpal et al., 2023) argues that linear mode connectivity can facilitate collaboration by merging models trained by different teams (Diskin et al., 2021) on various tasks; we show that DiLoCo is actually capable to do so *during* training, even when the data of each worker is different.

## 5. Limitations

Our work has several limitations, which constitute avenue for future work. First, we only considered a single task, namely language modeling, and a single architecture, a transformer. Other datasets, domains (*e.g.* vision), and other architectures (*e.g.*, CNNs which are known to be more sensitive to linear mode connectivity (Jordan et al., 2023)) should also be considered.

Second, we have presented results at the scale of 60 to 400 million parameters. However, at the time of writing state-of-the-art language models use 3 orders of magnitude more parameters. Therefore, it would be interesting to see how DiLoCo works at larger scale. Our initial extrapolation indicate that DiLoCo might perform even better at larger scales, because there is less interference during the outer-gradient averaging step. However, this hypothesis should be validated empirically.

Third, the version of DiLoCo presented here assumes that all workers are homogeneous. However, in practice workers might operate at wildly different speed. In these cases, waiting for all workers to perform the same number of steps is rather inefficient. Another avenue of future work is then to extend DiLoCo to the asynchronous setting, whereby workers update the global parameter without ever waiting for any other worker

Fourth, DiLoCo exhibits diminishing returns beyond 8 workers. Another avenue of future research is to improve the algorithm to better leverage any additional compute that might be available.

Finally, DiLoCo attains fast convergence in terms of wall-clock time. However, the distributed nature of the computation reduces the FLOP and data efficiency of the model, as shown by the $8\times$ updates row in Table 2. At a high level, this is because the outer updates have effectively too large a batch size; but naively reducing the outer-update batch size would result in the workers being destabilized because their batch-size is too small. Therefore, another avenue of future research is on balancing wall-clock time efficiency with compute efficiency and data efficiency, among other quantities of interest. In particular, we believe *asynchronous* variants of local-sgd may allow distributed training with relatively more data-efficient updates.

## 6. Conclusion

In this work we study the problem of how to distribute training of large-scale transformer language models when not all devices are co-located, and the network between the various machines may have low bandwidth. To address this problem, we propose DiLoCo, a variant of Federated Averaging whereby the outer optimizer is replaced with Nesterov momentum, the inner optimizer is AdamW (the *de facto* standard optimizer for transformer language models), and the number of inner optimization steps is large (our default value is 500). The latter is crucial to reduce communication, and it means that workers only need to send data once every 500 steps. Practically speaking, while standard mini-batch methods relying on data and model parallelism require sending data every few hundred milliseconds, DiLoCo does so only every few minutes. Therefore, if each communication step takes a lot of time, DiLoCo converges much faster in terms of wall-clock time.

Our empirical validation demonstrate the robustness of DiLoCo on several fronts, from the type of data distribution each worker consumes, to the number of inner optimization steps, and number of workers which can even change over time.

In conclusion, **DiLoCo is a robust and effective way to distribute training of transformer language models when there are several available machines but poorly connected**. Of course, it remains to be seen whether these findings generalize to models of larger scale, or to other domains and architecture types.

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

*Table 5.* **Optimization Hyperparameters** evaluated during in this work. Chosen values for main experiments are highlighted in bold.

| Hyperparameter | Value |
|---|---|
| Inner Learning rate | $4e^{-4}$ |
| Number of warmup steps | 1,000 |
| Weight decay | 0.1 |
| Batch Size | 512 |
| Sequence length | 1,024 |
| Outer Optimizer | SGD, SGDM, **Nesterov**, Adam |
| Inner Optimizer | AdamW |
| Outer SGD learning rate | 1.0, 0.7, **0.5**, 0.3, 0.1 |
| Outer SGDM learning rate | 1.0, 0.7, 0.5, **0.3**, 0.1 |
| Outer SGDM momentum | 0.9 |
| Outer Nesterov learning rate | 1.0, **0.7**, 0.5, 0.3, 0.1 |
| Outer Nesterov momentum | 0.95, **0.9**, 0.8 |
| Outer Adam learning rate | 1.0, 0.7, 0.5, **0.3**, 0.1 |
| Outer Adam beta1 | 0.9 |
| Outer Adam beta2 | 0.999, **0.95** |
| Outer Adam epsilon | 1.0, $\mathbf{10^{-1}}$, $10^{-3}$, $10^{-5}$, $10^{-7}$ |
| Communication frequency $H$ | 50, 100, 250, **500**, 1,000, 2,000 |
| Number of pretraining steps | 0, 12,000, **24,000**, 48,000 |
| Number of replicas | 4, **8**, 16, 64 |
| Data regimes | i.i.d., **non-i.i.d** |

# A. Appendix

## A.1. Implementation Details

**Hyperparameters**   We displayed in Table 1 the architectural difference between the 60M, 150M, and 400M models we evaluated. In Table 5, we outline the optimization hyperparameters considered for this study, and highlight in bold the values chosen for the main experiments. We detailled extensively the impact of each hyparameters in subsection 3.1.

**Inner Optimizer States**   In all experiments, the inner optimizer, `InnerOpt`, is AdamW (Loshchilov & Hutter, 2019) as standard practice when training transformer language models. Each replica in our method has a separate Adam state (*e.g.* first and second momentum). DiLoCo synchronizes the parameters of the model, but we also considered synchronizing the inner optimizer states. It did not lead to significant improvements while significantly increasing the communication cost ($\times 3$ more data to transmit). Therefore, we let each model replica own their own version of optimizer states. Similar findings were found in the literature where SGDM or Nesterov momentum were used as inner optimizers (Wang et al., 2020; Ortiz et al., 2021).

**Weighted Average of Outer Gradients**   In line 12 of Algorithm 1, we perform a uniform average of every model replica's outer gradient. There are also other strategies, such as *greedy soup* (Wortsman et al., 2022b) where model replicas are selected sequentially to minimize validation loss, or *disjoint merge* (Yadav et al., 2023) which uses a sign-based heuristics. The first strategy is too time-costly in our setting. We tried the latter, but got slightly worse results. Thus, for the random i.i.d. data regime we use a uniform average. For the non-i.i.d. data regime, we rescale each outer gradient by the number of examples in its shard. While at $k = 4$, all clusters are quite balanced, imbalance can be striking at $k = 64$ and giving more importance to larger clusters is beneficial.

**Infrastructure**   The empirical validation of this work was performed on machines hosting 16 A100 GPUs. These machines were not necessarily co-located in the same geographic region. The outer optimization step is performed on a CPU server connected to the local machines.

*Table 6.* **Pruning outer gradients** using a per-neuron sign pruning (Yadav et al., 2023).

| % of pruned values | Perplexity | Relative change |
|:---:|:---:|:---:|
| 0% | 15.02 | 0% |
| 25% | 15.01 | -0.06% |
| 50% | 15.08 | +0.39% |
| 75% | 15.27 | +1.66% |

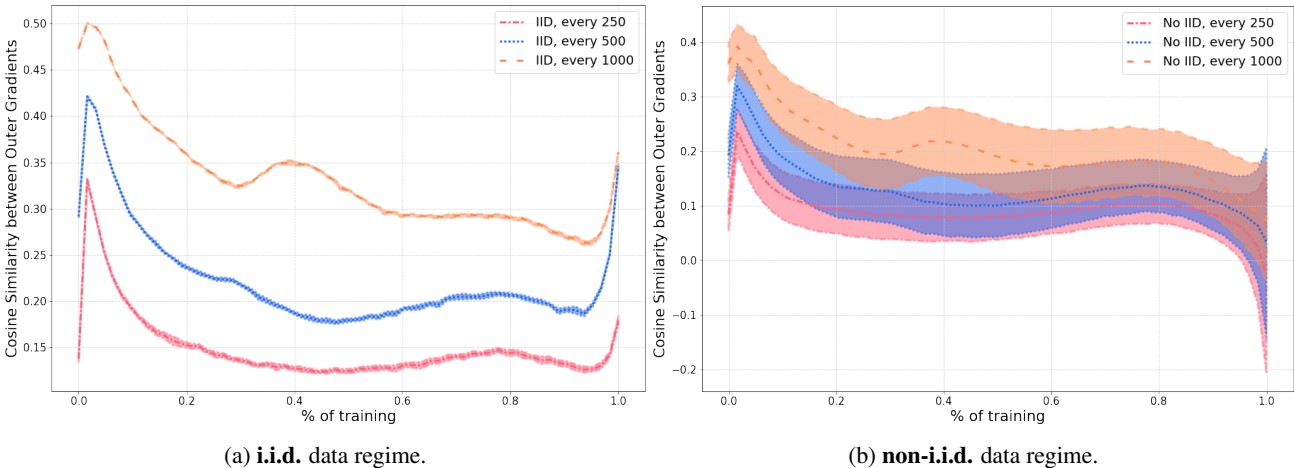

(a) **i.i.d.** data regime.          (b) **non-i.i.d.** data regime.

*Figure 8.* **Cosine Similarity between Outer Gradients**: The line is the average similarity among the $k = 8$ replicas' outer gradients, the shaded area is the standard deviation. This is almost null in the case of i.i.d. shards.

## A.2. Experiments & Ablations

**Pruning outer gradients**  Although DiLoCo communicates infrequently, when communication is required the network might get saturated, particularly when there are lots of workers, or when the model replicas are large. We thus explored pruning of outer gradients in order to reduce the need for high-bandwidth networks.

We consider the simplest pruning technique, sign-based pruning following Yadav et al. (2023). More efficient methods could be explored in the future (Tang et al., 2023), particularly those leveraging structured sparsity. In Table 6, we prune between 25% to 75% of the individual outer gradients per replica before averaging them. Pruning up to 50% of the individual values resulted in negligible loss of performance ($+0.39\%$ perplexity). Therefore, DiLoCo's communication efficiency can be further improved using standard compression techniques.

**Cosine Similarity of outer gradients**  Our empirical validation shows remarkable robustness of DiLoCo to the number of inner optimization steps and data distribution of the shards. Why does DiLoCo converge even when performing 500 inner steps? And why using shards with different data distribution does not harm performance at all?

To shed light on these questions, we have gathered statistics of the outer gradients returned by workers. In particular, we calculate the average cosine similarity between outer gradients returned by workers while varying the number of inner optimization steps ($H = \{250, 500, 1000\}$) for both the i.i.d. (in Figure 8a) and non-i.i.d. (in Figure 8b) settings.

The former regime has close to no variance compared to the latter, since all shards have the same data distribution and therefore outer-gradients are much more correlated. For both data regimes, perhaps unintuitively, similarity is inversely proportional to the communication frequency however. We surmise that when the number of inner step is larger (up to some extent) model replicas converge towards a similar general direction (Gu et al., 2023) averaging out the noise of stochastic gradient descent.

Interestingly, as the learning rate anneals to 0 towards the end of training, the outer gradients similarity increases in the i.i.d.

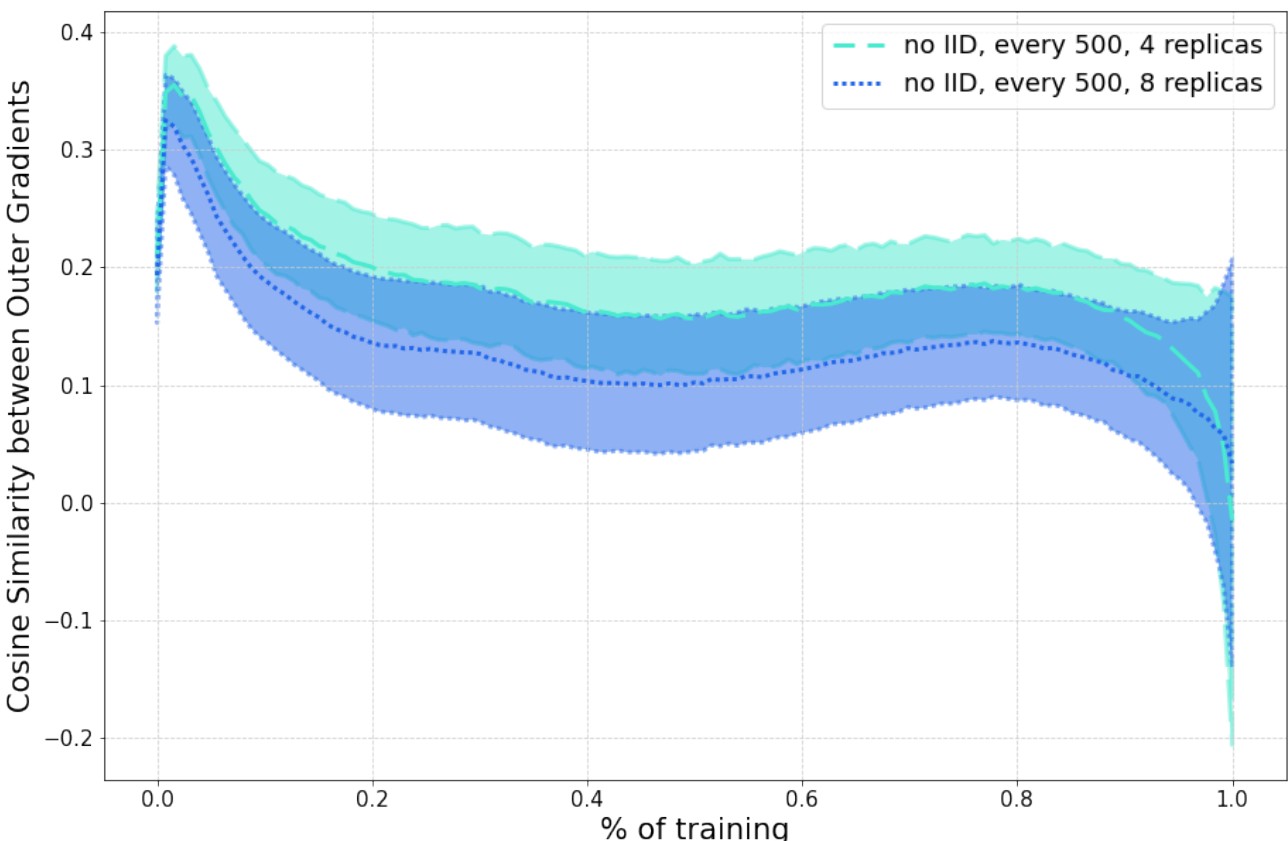

*Figure 9.* **Outer Gradients similarity versus number of replicas**: in a non-i.i.d. data regime increasing the number of replicas/clusters ($k = 4 \rightarrow 8$) produces more dissimilar outer gradients.

case but in the non-i.i.d. case only the variance increases. Since shards have a different distribution, each local optimization seems to fall in a different nearby loss basin. However, the averaging of such more orthogonal gradients grants beneficial generalization as the non-i.i.d. version of DiLoCo tends to generalize at a better rate towards the end of training as can be seen in Figure 5.

Lastly, in the non-i.i.d. setting we expect that the larger the number of shards the more distinctive their distribution, and therefore, the less correlated the corresponding outer gradients. Figure 9 shows precisely this trend when going from $k = 4$ to $k = 8$ shards. We also found that in this setting the averaged outer gradient's norm is inversely proportional to the square root of the number of replicas.

**Asynchronous Communication** In DiLoCo all workers communicate their outer-gradients after $H$ inner optimization steps. In practice, it might happen that a worker gets rebooted or that the network might lose packets. In these cases, communication is not feasible.

In Figure 10, we simulate such inability to communicate by randomly dropping outer gradients with probability equal to 0% (in teal), 10% (in purple), 30% (in red), to 50% (in orange). When an outer gradient is dropped, the local worker continues training for the following $H$ steps starting from its own parameters $\theta_i^{(t)}$ (as opposed to the shared parameters $\theta^{(t)}$).

In both i.i.d. and non-i.i.d. data settings, a higher probability of being dropped results in more unstable learning with transient spikes in perplexity. However, even in the extreme non-i.i.d setting where each replica has 50% probability of dropping communication, the degradation of perplexity relative to perfect communication is only 2.1%. Consequently, with robustness to communication failure, the need of a synchronization barrier is less critical and thus training can be accelerated without having to wait all replicas.

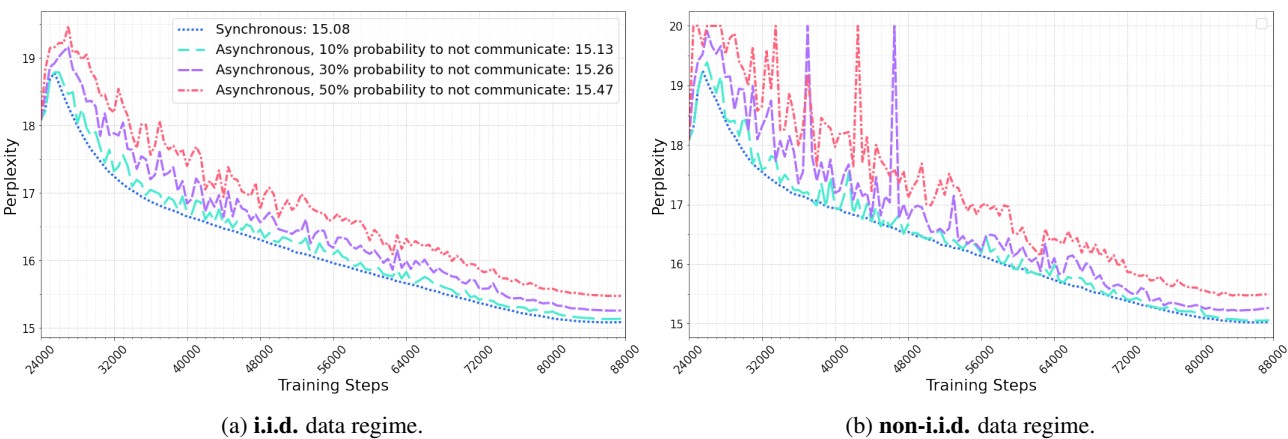

(a) **i.i.d.** data regime.        (b) **non-i.i.d.** data regime.

*Figure 10.* **Asynchronous communication**: we drop communication of outer gradients of each replica with a certain probability. If a replica is dropped, it continues training without synchronizing its parameters.

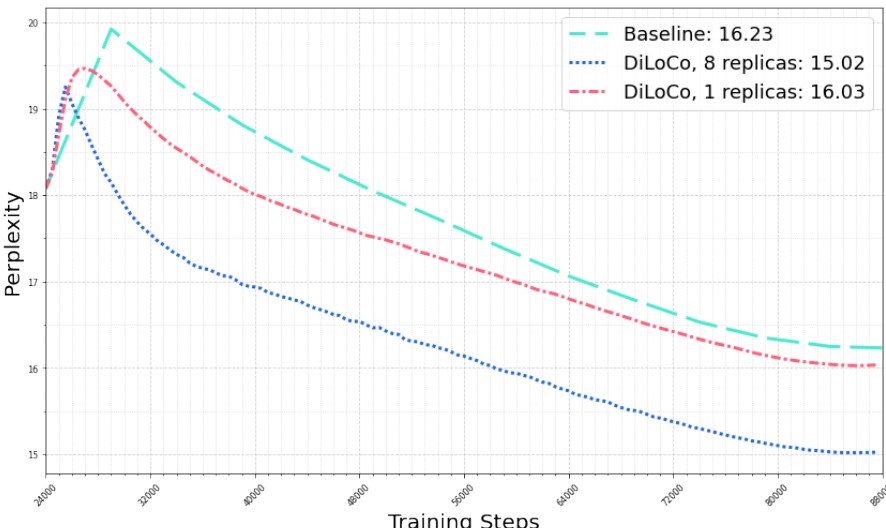

*Figure 11.* **Accelerating a single worker**: DiLoCo applied to a single replica $k = 1$ provides both faster and better generalization.

**Accelerating a single worker**    Recently works have shown that linear connectivity can also *accelerate* the convergence of a non-distributed model. For instance, the Lookahead optimizer (Zhang et al., 2019) proposes to take an outer step of outer SGD using a single replica, which is equivalent at interpolating between the starting and end points of a phase in the parameters space.

Figure 11 shows that DiLoCo applied to a *single* replica/cluster ($k = 1$ but $H \gg 1$) can improve both convergence speed and final generalization performance at null communication cost. Specifically, every $H = 500$ inner steps, we compute the only outer gradient as described in Algorithm 1, and then (locally) update the parameters using the outer optimizer. This experiment further corroborates the robustness and wide applicability of DiLoCo.

