# OpenReview forum: "DiLoCo: Distributed Low-Communication Training of Language Models"
_ICML.cc/2024/Workshop/WANT — WANT@ICML 2024 Poster_

### Official Review · Reviewer_NHxt · 2024-06-11
**Review on DiLoCo**

**Confidence:** 4

**Summary:**

A distributed federated learning strategy, DiLoCo is presented, which achieves high accuracy on LLM training with low communication overhead. In general, the presented concept could be useful for exploiting federated learning infrastructures. But there are several weaknesses in the work which are elaborated below. Primarily, it seems that there is limited novelty in the paper, given that the method is adopted from an already established algorithm. However it is still interesting to see the application of this methodology for an LLM use-case, but it seems that (1) the considered models are quite small and (2) heterogeneity is not explored. If the authors can extend their study to include these two points in a revised manuscript, it could be interesting to see the results. Furthermore, profiling statistics and information on the systems is missing, which makes it difficult to evaluate the performance.

**Strengths:**

- Achieves relatively better accuracy with low communication over other investigated benchmarks
- Extensive parametric analysis
- Potential for exploitation of federated learning

**Weaknesses:**

The authors are encouraged to work on the suggestions here:

- [line 39] the inner optimizer is replaced with AdamW -- What does AdamW replace?

- [line 44] 'workers need not to communicate at each and every single step but only every H steps which can be in the order of hundreds or even thousands' -- Have you tested gradient accumulation? How is this approach different compared to gradient accumulation, except the heterogeneous architecture possibilities? From the conclusion, it seems heterogeneity is indeed not explored here.

- Although the authors talk about heterogeneous architectures, this is not explored in the paper. This is also mentioned later as a limitation. This is quite a critical limitation, since then the presented methodology could very well be performed with standard gradient accumulation approach.

- The considered model sizes are rather small, given the state of the art in the field. Can the authors provide some results with more parameters? Since this will significantly increase the complexity, the values of H and T in this case will perhaps be quite critical. This is mentioned in the conclusion too, however this is a crucial point for this paper, since the title of the paper is on LLMs.

- Why only consider perplexity? It is not clear how this term is defined and whether there is some kind of normalisation involved. At the moment, the reported perplexity values seem quite close to each other. Hence it is difficult to say if the improvement over benchmarks is indeed significant.

- In general it is difficult to follow what the baselines are? Suggestion is to add a table with all baselines (Baseline 1, 2, etc.) along with information if they are data parallel or serial and with pre-training or not and then use these numbers consistently throughout the document.

- Table 2: The information in this table is rather ambiguous. What does 'Baseline, 8x updates' mean? What is 'microbatching' in this table? I am not sure if this is the correct word here, since I think the authors mean a smaller batch size? If it is indeed smaller batch size, that would mean there is lower GPU utilisation. Have the authors investigated parameters, such as GPU utilisation? In general, such profiling statistics should be added to this investigation.
Also the values in this table assumes the performance to be ideal (1x, 8x etc.). However in practise, this is hardware-dependent. The authors should provide a background on the systems and provide the real compute/communication times obtained through profiling the code.

- Table 3: It is rather odd that the accuracy is not influenced to large extent by changing the number of workers. This could be an influence of the model itself. The authors talk about three models used in this study in Tab. 1. Which of these configurations is used for this result? It would be informative to see the effect of this with even larger model.

- Table 4: What are Relative (%) and Absolute (PPL)? - Please define these quantities.

- [line 294] Larger models are more efficient at fitting the same amount of data - What is efficiency here?

- Conclusion: It is not clear where the "low bandwidth" is addressed in this paper. Authors should write about the bandwidth under which the results are obtained, and how this impacts the comparison to other approaches.

---

### Official Review · Reviewer_zkbr · 2024-06-12
**DiLoCo: Distributed Low-Communication Training of Language Models**

**Confidence:** 4

**Summary:**

This paper focuses on designing a new federated learning mechanism, DiLoCo, to reduce the communication cost. The idea is that clients locally update their model using their own data. Then, after the H steps, local models are aggregated. This process continues until convergence.

**Strengths:**

+ The research direction, which focuses on reducing communication costs, is interesting.

**Weaknesses:**

- Compared to the existing work, the paper's contribution is unclear. Similar work in the context of local SGD [1] and in federated learning setup [2] have been widely studied in the literature.

[1] Stich, Sebastian U. "Local SGD converges fast and communicates little." arXiv preprint arXiv:1805.09767 (2018).
[2] Gao, Hongchang, An Xu, and Heng Huang. "On the convergence of communication-efficient local SGD for federated learning." In Proceedings of the AAAI Conference on Artificial Intelligence, vol. 35, no. 9, pp. 7510-7518. 2021.
....

- The contribution of using Nesterov as the outer optimizer, as suggested by the authors, is incremental.

- The experimental results are limited as only one model and dataset have been considered.

---

### Official Review · Reviewer_xYuj · 2024-06-13
**Promising technique for training large models on poorly connected clusters of accelarators**

**Confidence:** 4

**Summary:**

The authors proposed a distributed optimization algorithm (DiLoCo) to train language models on various islands of devices which do not have high-bandwidth inter-connectivity. The method is similar to traditional data parallelism with limited communication. The computational resources are split in $k$ workers or islands. Each island corresponds to a set of closely connected accelerators. Data are split in $k$ shards, so to have a shard per worker. Each worker receives a replica of the model, which is trained for $H$ optimization steps before synchronizing with the other workers. The fact that $H\gg1$ alleviates the need for high-bandwidth communication between the workers.

The authors test their approach by training decoder-only transformers adapted from the Chinchilla architecture on C4 dataset. They also provide an extensive ablation study. The results show that the method outperforms baselines when measuring the perplexity of the model. Baselines consist in increasing the batch size or increasing the number of optimization steps. The method is also reported to converge faster in terms of overall training time.

The paper presents a simple, elegant, and promising method to help with the training of large language models in low-bandwidth communication settings. It is well aligned with the workshop topics. I recommend for accepting the paper.

**Strengths:**

- The paper is clear, well organized and written;
- The idea is simple and easy to follow;
- Many experiments investigate different points of the approach (communication frequency, data regimes, number of replicas);
- The framework is elastic and can adapt to variation in the number of workers;
- Authors are aware of some important limitations.

**Weaknesses:**

Even if it is already mentioned in the limitations by the authors, the fact that DiLoCo handles heterogeneous workers would be indeed much welcome. If one worker goes much faster than the others, it could remain idle most of the training.

It might be a point that I missed, but it would be interesting to compare DiLoCo with traditional data parallelism, especially in terms of latency and overall training time.

The core idea of DiLoCo is to enable training on low-bandwidth communication. It would be valuable to provide a lower bound, even rough one, about the communication bandwidth respective to the model size. Indeed, is it possible that the model may be so large, and communication so slow that outer steps take too much time for a realistic training. Providing figures would help appreciating the usefulness of DiLoCo in real settings.

More generally regarding network, what are the requirements for DiLoCo? Is the connection between workers on different location totally transparent to DiLoCo? Otherwise, how are network issues mitigated (like lost packets or dropped connections)?

---

### Meta-Review · Area_Chair_Mbnf · 2024-06-17

**Recommendation:** Reject
**Confidence:** 4

**Metareview:**

The work proposes a method for distributed optimization that utilizes a version of Federated Averaging with the Nesterov momentum as the outer optimizer. Reviewers have appreciated the potential of the research idea, as well as the depth of the empirical analysis of the proposed method. However, they also expressed strong concerns related to the novelty of the approach, the lack of clarity around systems-level data about evaluation, such as the network bandwidth requirements. Furthermore, two reviewers noted that the method is not explored in the context of heterogeneous setups, which makes the contribution of the paper less clear given the existence of gradient accumulation.

Therefore, I cannot recommend this paper for acceptance, but I encourage the authors to revise the manuscript by following the reviewers' suggestions and submit it to another venue.

---

### Decision · Program_Chairs · 2024-06-18

**Decision:**

Accept (Poster)

**Comment:**

After a thorough evaluation of the paper and the feedback provided by reviewers and a meta-reviewer, we have made the decision to accept the paper. Our decision is motivated by the paper's relevance to one of the core topics of the workshop. The program chairs believe that the paper presents valuable results that are pertinent to the workshop's objectives, and we are eager to foster discussions and research advancements in this area. Please take into account all reviewers' suggested improvements. Congratulations and hope to see you in person at the workshop and brainstorm on efficient training research together!